# Toward Cost-Effective Mobile Video Streaming through Environment-Aware Watching State Prediction

**DOI:** 10.3390/s19173654

**Published:** 2019-08-22

**Authors:** Xuanyu Wang, Weizhan Zhang, Xiang Gao, Jingyi Wang, Haipeng Du, Qinghua Zheng

**Affiliations:** MOEKLINNS Lab, School of Computer Science and Technology, Xi’an Jiaotong University, Xi’an 710049, China

**Keywords:** sensors in mobile phones, cost effective, mobile video streaming, sensor-based environment-awareness, user behavior, watching state prediction

## Abstract

Mobile video applications are becoming increasingly prevalent and enriching the way people learn and are entertained. However, on mobile terminals with inherently limited resources, mobile video streaming services consume too much energy and bandwidth, which is an urgent problem to solve. At present, research on cost-effective mobile video streaming typically focuses on the management of data transmission. Among such studies, some new approaches consider the user’s behavior to further optimize data transmission. However, these studies have not adequately discussed the specific impact of the physical environment on user behavior. Therefore, this paper takes into account the environment-aware watching state and proposes a cost-effective mobile video streaming scheme to reduce power consumption and mobile data usage. First, the watching state is predicted by machine learning based on user behavior and the physical environment during a given time window. Second, based on the resulting prediction, a downloading algorithm is introduced based on the user equipment (UE) running mode in the LTE system and the VLC player. Finally, according to the corresponding experimental results obtained in a real-world environment, the proposed approach, compared to its benchmarks, effectively reduces the data usage (14.4% lower than that of energy-aware, on average) and power consumption (about 19% when there are screen touches) of mobile devices.

## 1. Introduction

In recent years, an increasing number of video applications involving entertainment, news, instructional videos, etc. have been developed for mobile users. In order to achieve high-quality media content streaming on the Internet, researchers have produced studies regarding the delivery level, among which HTTP adaptive streaming (HAS) occupies a dominant position, as its flexible service model allows users to increase or decrease video quality as needed during playback [1]. For example, YouTube has adopted HAS to bring adaptive streaming to mobile devices and TVs [2]. In addition to the delivery level, however, at the data download level, to reduce resource consumption, there are two major problems with mobile video applications. First, downloading video data consumes a substantial amount of power. Second, once enough data have been downloaded, users’ skip/quit operations will lead to downloaded data being unused. Even though HAS is so commonly applied in video delivery, these two problems still exist on the data download level, for which there are the following two main reasons. First, the client may have the capacity to download extra video segments when the bandwidth provided to users is larger than the highest bitrate of HAS. Second, the k-push strategy in HAS over HTTP/2 leads to the download of segments that may be unused.

Therefore, setting our sights on the data download level, this paper focuses on data download strategies to provide cost-effective video streaming. Several studies have already examined cost-effective mobile video streaming, focusing on providing bandwidth or energy saving schemes [3,4,5,6,7,8,9]. Specifically, some researchers considered user behavior and proposed optimized cost-effective strategies [10,11,12]. However, these studies did not consider the impact of the physical environment on user behavior. Thus, the optimization methods proposed above could fail in diverse physical mobile environments since a classic study has already shown that user behavior is affected by the physical environment [13].

Therefore, in this paper, we propose a cost-effective mobile video streaming scheme by considering both the physical environment and the operation behavior of a mobile user. First, a prediction model of the user watching state is developed utilizing machine learning methods, revealing the influence of the user’s physical environment on the user’s operation behavior. An actual watch record from an online education platform is used for the case study in the experiment. Second, a downloading algorithm is proposed to provide differentiated download strategies according to stable and unstable user watching states. When the state is stable, the video streaming strategy tries to save energy through data transmission batching. When the state is unstable, the video streaming strategy seeks to minimize the download of unused data via a conservative download method. Third, we conduct verification experiments to verify the advancement of the algorithm, and the experimental results show that the proposed scheme, compared to existing methods, can effectively reduce power consumption and mobile data usage. Finally, to implement our algorithm, a mobile video streaming prototype based on the VLC player is developed on the Android platform. The main contributions of this paper are summarized as follows.
The proposed scheme does not concentrate on deploying a specific delivery system for video streaming. It provides a cost-effective data download strategy through environment-aware watching state prediction and provides a generalized strategy that can be used for many other video delivery technologies, extending the domain of the traditional data download algorithms for mobile videos.The proposed data download algorithm considers both the physical environment and the operation behavior of a mobile user. Machine learning is adopted to reveal the influence of the user’s physical environment on the user’s operation behavior. In this manner, the data download algorithm can provide differentiated download strategies according to the watching state of the user.

The rest of this paper is organized as follows. In Section 2, related work is presented. Section 3 introduces user watching state prediction based on machine learning. Section 4 presents the download algorithm based on the watching state. Section 5 describes the implementation and evaluates the performance of our solution. Section 6 concludes the paper.

## 2. Related Work

To reduce bandwidth and computing resource consumption on mobile devices, work has been done from several perspectives. First, dynamically adjusting the size of the buffer is one solution. Exploring the impact of the mobile cache threshold on video streams, Wu et al. [3] proposed an adaptive cache threshold algorithm to achieve cost-effectiveness and reduce resource consumption of unused content. Moreover, Ghoreishi et al. [4] proposed using a hierarchical cache structure to determine the optimal video buffer size at different layers, thus minimizing the ratio of transmission bandwidth cost to storage cost. For the LTE core network, Zhu et al. [7] proposed using a cache framework in the network to optimize data requests. By analyzing the energy efficiency of DASH (Dynamic Adaptive Streaming over HTTP) in the LTE network environment and the energy consumption of different fluidization strategies (segment length and buffer size), the ratio resource control (RRC) was analyzed, and its impact on energy consumption was quantified to reduce energy consumption in Zhang et al. [11]. Furthermore, MEC can help enhance adaptive bitrate (ABR) video delivery by combining content caching and ABR streaming together [9].

Second, solutions from other perspectives can be applied for some specific tasks. Video codecs can be optimized to reduce power consumption during encoding, decoding, and display [5]. For mobile tasks that consume huge amounts of power such as illumination, Liu et al. [6] presented content-adaptive display (CAD), to transfer the load of luminance compensation for videos from the mobile device’s CPU to the GPU to produce power savings. A decision algorithm to select the most efficient video codec according to its chroma characteristics was introduced by Jiménez et al. [14] to save bandwidth.

Third, novel and user-friendly changes can be made considering the influence of user behavior. To reduce the resource consumption of mobile devices, Hu W. et al. [10] analyzed the power consumption mode of the LTE wireless interface and dynamically adjusted the data transmission algorithm based on the user’s touch screen behavior to keep the LTE interface in the energy-saving state for as long as possible. By analyzing the relationship between the time span of video viewing and the power consumption of mobile devices, Li et al. [12] adjusted when and how data were downloaded according to the viewing timespan to reduce power consumption. Aguiar [15], Brinton [16,17], and Sinha [18] classified user participation and learning mode by analyzing the click stream. Yang et al. [19] used the user’s click stream in a Markov model to predict the user’s next click. The leaving time was modeled to predict which users tended to leave prematurely and permanently using machine learning algorithms [20]. Machine learning algorithms are used to analyze learners behavior and detect student withdrawal at an early time in the course based on MOOC (massive open online courses) datasets [21].

In the above research on mining and predicting user behavior to achieve power conservation for mobile video applications, the sequence of historical operations that occur when a user watches a video is used as the basis for prediction and further saves energy in mobile devices, but the influence of the physical environment on user behavior is not taken into account, and the consideration of factors affecting user behavior is not comprehensive or accurate enough, resulting in low prediction accuracy. Therefore, in this study, the influence of the user’s environment is considered, and the scope of user behavior prediction is expanded to the prediction of the user’s viewing status rather than specific operations.

## 3. User Watching State Prediction

During video playback, users perform various operations such as starting playback, pausing, skipping, and quitting. These operations may result in downloading unused data. For example, a portion of the video data may have been downloaded into RAM only to be skipped by the user. In practice, a user’s operation behavior is affected by various factors, so it is difficult to obtain accurate predictions. Therefore, the task of predicting the user’s operation behavior is transformed into that of predicting the watching state in this study, and such states can be divided into two categories: (1) **stable**, where the user continues watching the video within the prescribed time interval *T*, and (2) **unstable**, where skip/quit operations occur within the prescribed time interval *T*. The reason why we divided the user watching state into only two categories is that, since the criterion for this classification is stability, besides stable and unstable, there should be at most one intermediate state. However, the main purpose of classifying the user watching state is to determine which downloading strategy out of two should be applied under certain circumstances, and further achieve cost effectiveness. The two alternative strategies achieve minimum data waste and minimum power consumption respectively, which are two representative characteristics of cost effectiveness. Therefore, in order to address these two strategies one by one, only two user watching states need to be defined.

In this study, we analyze the behavior of a user watching an educational courseware video, and three factors are considered to predict the user’s watching behavior: (1) courseware video information, (2) physical environment, and (3) historical watching behavior. These three factors are analyzed to obtain effective features, and machine learning is used to model and predict the user’s next watching state. We collected video content information, sensor data, and learning operation logs from an online educational platform from 1 September 2016–31 December 2017 (486 days), including data for 3905 students and 380 courses (6164 courseware videos). In total, 57,207 samples were obtained through the video player to train the model, and the dataset is made available to potentially further benefit the community since it may be useful for other environment-aware studies [22]. Note that there are many types of video content, such as news, entertainment, and education. Users’ response characteristics vary with content type. However, the research methodology based on these educational videos can also be reasonably applied to other video-streaming applications such as movies or videos just for pleasure.

### 3.1. Courseware Video Information

The data for a courseware video include course ID, courseware ID, and duration. Duration is recorded because, in addition to video content, it affects the user’s watching time. Statistics for the learning time obtained from learning operation logs of the online education dataset are shown in Figure 1. The left part of the figure shows that when the video duration is greater than six minutes, the average watching time remained less than 10 minutes. The right part of Figure 1 indicates that the average ratio of watching time to video duration continued to decrease with the increase of video duration.

### 3.2. Physical Environment

To determine the impact of the physical environment on users’ behavior, it is necessary to assess the users’ current physical environment. Therefore, this study determines whether the current environment is noisy or quiet based on the sensors in the mobile client. Guided by a previous study [23], acceleration sensors, a microphone, and features extracted from those data (mean, standard deviation, median, skewness, kurtosis, and quartile range) are used in this paper to examine the physical environment, all given equal weights.

As a user watches a video, sensor data are collected at a frequency of 1 Hz and saved as a CSV file with a unique identifier. The acceleration sensor’s output includes values in three directions ($AAC_{X}$, $ACC_{Y}$, and $ACC_{Z}$). The variable ACC, defined as $ACC=\sqrt{AAC_{X}^{2}+ACC_{Y}^{2}+ACC_{Z}^{2}}$, is added as a uniform measure. The microphone captures the sound intensity of the current environment. The volume of sounds in real life is described in decibels, but the physical quantity provided by Android is amplitude. Hence, we convert the raw data from amplitude to decibels.

Because it is difficult for the original data to reflect directly the state of the physical environment, we used a sliding window to extract six features—mean, standard deviation, median, skewness, kurtosis, and quartile range—which were proven effective in an existing study [23]. Skewness and kurtosis describe the steepness and symmetry of a distribution, respectively. These statistics were compared with those of a Gaussian distribution.

### 3.3. Operation Behavior

When a user watches a video, he or she may engage in various click operation behaviors. We recorded operations including playback, pausing, skipping, and quitting, as well as the duration of each operation, to represent the entire video watching process. The features we extracted for operational behavior included cumulative watching time, pause time, drag time, number of playbacks, number of pauses, number of drags, playback ratio, pause ratio, drag ratio, and semantic weight, which were obtained from the user operation sequence using two-level analysis and quantification.

**Level 1: Serialization.** First, the click operations were coded and serialized. The records of a users’ playback, quitting, pausing, and dragging operations are represented by 1, 2, 3, and 4, respectively. For example, the sequence of operations in Figure 2 is serialized as “1314432”.

**Level 2: Weighting behavior patterns.** Second, we defined two semantics: stable and unstable. Next, we used fuzzy string matching to calculate the semantic weight of the operation sequence. Each semantic contained multiple substrings of length *n*. According to the study of [18], we counted substrings of length four (called a *pattern*) in all operation sequences, selected the top 50 most frequent substrings, and assigned semantics to them, as shown in Table 1. Then, we calculated the weights of the stable and unstable semantics for each sequence of operations. The semantic weight of the *i*th operation sequence sample $s_{i}$ in semantic *j* was calculated by the formula $w_{ij}=\sum_{p=1}^{t_{j}}Weight(P_{jp},s_{i})$, where $P_{jp}$ is the *p*th pattern in semantic *j*, $t_{j}$ is the total number of patterns in semantic *j*, and $Weight(P_{jp},s_{i})$ calculates the weight of sample $s_{i}$ in the *p*th pattern in semantic *j*. There is no need for exactly four operations since for shorter videos, by the time the user performs enough operations, the video may already be nearly over. We only need to know how many times the four-digit patterns occurred during the time window. If none of those patterns occurred, there were other variables in the input without the weighted behavior patterns.

### 3.4. Watching State Prediction

We selected the seven most commonly-used machine learning algorithms, namely ridge, LASSO, elastic net, ExtraTrees, random forest, gradient boosting, and XGBoost, for modeling and prediction of the watching state. The 10-fold cross-validation method was used to prevent overfitting. The evaluation indexes of the results were modeling time, mean absolute error (MAE), root mean squared error (RMSE), and the coefficient of determination $R^{2}$. MAE can avoid the problem of mutual cancellation of errors, so it can accurately reflect the actual prediction error. RMSE measures the deviation between the observed value and the true value. The smaller MAE and RMSE are, the higher the quality of the prediction is. $R^{2}$ determines the degree of closeness. An $R^{2}$ closer to one represents a higher quality prediction model. The results are shown in Table 2. We observe that, except for the longest modeling time of XGBoost, modeling times of the remaining algorithms were relatively short. Additionally, we observe that the $R^{2}$ of random forest was the highest, with the minimum MAE and RMSE at the same time (0.0093 and 0.0114, respectively). Therefore, we used random forest to predict the user watching state. The time consumed by the prediction was short enough to support real-time delivery.

## 4. Downloading Algorithm Based on the Watching State

Since the LTE technology dominates the current mobile telecom infrastructure, downloading was considered to use the LTE network as the platform for video transmission. In the LTE system, the user equipment (UE) has two running modes that are switched by the RRC by changing the state of the wireless interface. These modes are LTE-ACTIVEand LTE-IDLE, and during the switch from LTE-ACTIVE mode to LTE-IDLE mode, there is an intermediate high-power state of duration $t_{tail}$, called TAIL. When the system is transmitting data, the UE is in high-power LTE-ACTIVE mode. When data transmission ends, the UE enters low-power LTE-IDLE mode after the $t_{tail}$ TAIL.

To both save power and reduce the waste of mobile data, the idea of our algorithm is to make a decision based on the user’s watching state during time window *T*. When the user is in the stable state, we can fully utilize the client buffer to download enough data in the LTE-ACTIVE mode and then switch to LTE-IDLE and remain in that state as long as possible. When the user is in the unstable state, we can download a group of pictures (GoP) that the user will watch, minimizing the download of unused data. A GoP contains a fixed number of *I*, *P*, and *B* frames and is the basic unit of video transmission, with a duration of $l_{i}$ and a data size of $d_{i}$.

At present, there are two typical cost-effective mobile video streaming scheduling strategies. (1) bitrate streaming [24] is a strategy that downloads a GoP only when it needs to be played back. The advantage of this strategy is that it minimizes the amount of unused data being downloaded and the required buffer space. However, network fluctuations may cause video stalling. (2) ON-OFF streaming [25] is a strategy implemented by exploiting the buffer capacity of the mobile client. It seeks to download video data into the buffer at the highest speed until the buffer is filled; afterwards, it switches the LTE mode into a power-saving state. When the data size in the buffer is less than the threshold, data downloading is resumed.

If the video continues to play during time window *T*, the data size is $D_{T}=v*T$ during this period, where *v* is the video playback rate. For bitrate streaming, since the UE is always in the LTE-ACTIVE mode, the total energy consumed during time window *T* is calculated by Equation (Equation 1). For ON-OFF streaming, multiple GoPs are downloaded in the LTE-ACTIVE mode before switching the wireless interface to the TAIL state. The energy consumption of downloading the *i*th GoP under this strategy is calculated in Equation (Equation 2) by an existing method [10], where Δt is the time interval for downloading $g_{i}$ and $g_{i-1}$. The power consumption values of the TAIL, ACTIVE, and PROMOTION (the process of switching from IDLE to ACTIVE) states are expressed as $P_{tail}$, $P_{active}$, and $P_{pro}$, respectively. At the same time, the data throughput in the network is *r*.

(1)$$E_{T-Bitrate}=T*P_{active}$$

(2)$$E_{i}=\left\{\begin{array}{c} P_{pro}*t_{pro}+P_{active}*\frac{d_{i}}{r}+P_{tail}*t_{tail},if\;\Delta t>t_{tail}\\ P_{active}*\frac{d_{i}}{r}+P_{tail}*t_{tail},Otherwise \end{array}\right.$$

The total energy consumption of downloading *k* GoPs during a given time window *T* is $E_{T-ON-OFF}=\sum_{j=i}^{i+k}E_{j}$. If each GoP is downloaded, the UE running mode will be switched to LTE-IDLE, and the maximum amount of energy will be consumed during the window, as shown in Equation (Equation 3). For the minimum value described in Equation (Equation 4), the UE running mode will remain in the LTE-IDLE mode for as long as possible, reducing the number of promotions and leading to the minimum energy consumption during the time window.

(3)$$E_{T-ON-OFF}^{max}=k*\left(P_{pro}*t_{pro}+P_{tail}*t_{tail}\right)+\frac{D_{T}}{r}*P_{active}$$

(4)$$E_{T-ON-OFF}^{min}=minimize\sum_{j=i}^{i+k}E_{j}$$

**Calculating the duration of time window *T*.** To save energy, ON-OFF streaming needs to be adopted, but there will be additional energy consumption due to state switching. When using bitrate streaming, since the amount of data downloaded in the beginning is small and there is no state switching, power consumption is low. However, there is a threshold that makes the power consumption of bitrate streaming higher than that of ON-OFF streaming. Therefore, our objective is to determine the minimum time window in Equation (Equation 5). Combining Equations (Equation 1) and (Equation 3), we can obtain Equation (Equation 6). By letting *c* be defined as Equation (Equation 7), we can obtain Equations (Equation 8) and (Equation 9). If a state transition occurs, there must be more than two GoPs downloaded in a time window, i.e., k>2. The size of the time window *T* obtained here serves as the basis for the subsequent analysis of the learning behavior and physical environment in Section 3.

(5)$$E_{T-Bitrate}>E_{T-ON-OFF}^{max}$$

(6)$$T>k*\frac{r}{r-v}*\frac{P_{pro}*t_{pro}+P_{tail}*t_{tail}}{P_{active}}$$

(7)$$c=\frac{P_{pro}*t_{pro}+P_{tail}*t_{tail}}{P_{active}}$$

(8)$$T>k*\frac{r}{r-v}*c$$

(9)$$T_{min}=k*\frac{r}{r-v}*c$$

**Determining the mode switching time of ON-OFF streaming in the stable state.** We used the lower bound β and the upper bound α of the buffer to decide when to start and stop downloading. When the user is in the stable state, if $D_{T}<B$, $\alpha =D_{T}$, and the wireless interface is activated to download the amount $D_{T}$ of data into the buffer. There is no state switching during the download process, so there is no need to determine the lower bound β. If $D_{T}<B$, α=B, and according to [10], the lower bound β can be calculated by $\frac{r}{2v-r}B$, where the probability of the occurrence of skip/quit operations is $(1-r^{2})p$, where $r^{2}$ is the accuracy of the prediction result and *p* is the probability of skipping/quitting.

If the predicted result of the watching state is the unstable state, we switch the strategy to bitrate streaming to minimize the download of unused data. Algorithm 1 describes the downloading algorithm, where the function predict() returns a Boolean value (0,1), indicating either a stable or an unstable state. 

**Algorithm 1:** Cost-effective mobile video streaming downloading algorithm with environment-aware watching state prediction.

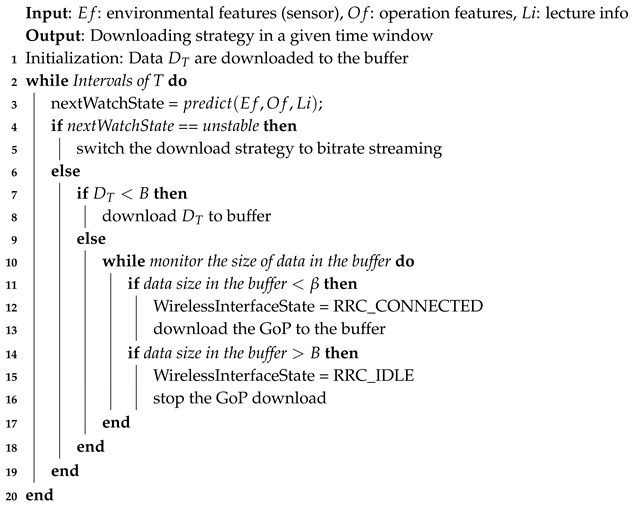



The proposed algorithm in this section can switch the download strategy between bitrate streaming and ON-OFF streaming and determine the mode switching time of ON-OFF streaming. However, it relies heavily on the prediction result of the user watching state in Section 3, which makes it error-prone when the prediction is delayed and may fluctuate when new factors occur, resulting in extra data waste or power consumption.

## 5. Experiments

After introducing the above video data downloading algorithm in Section 4, in this section, we apply it on the Android application platform and compare it with two representative approaches, bitrate streaming [24] and the energy-aware algorithm [10], to verify the usability and effectiveness of our proposed algorithm, proving that the application of the algorithm in the system can improve the endurance of the UE and save data traffic costs.

### 5.1. Experimental Setup

The proposed strategy was developed in an Android application that uses the VLC player by modifying the stream_output and user_interfaces modules. The default download strategy for VLC is bitrate streaming [24]; we added the energy-aware approach [10] and our solution to the stream_output module and added a strategy switch interface to the user_interfaces, which was controlled by the predicted watching state. Afterwards, we compiled and generated libvlc.aar and embedded it into our application. ON-OFF streaming was used in both our strategy and the energy-aware method, where the default buffer size is 10 MB in VLC. We used MediaPlayer.Event to intercept the user’s click operation and SensorManager provided by Android to collect sensor data at the frequency of 1 Hz. The data were stored in memory, and the features were generated locally and sent to the predictive model.

In the experiment, an Android cell phone was used as our client, and third-party applications were used to measure throughput and power consumption; power consumption was measured in mAh, and voltage was assumed to be 3.7 V during the experiment.

### 5.2. Experimental Results and Analysis

We analyzed the performance of our solution by comparing it with existing algorithms. We used three different algorithms to record the total energy consumption and the unused data under different conditions. The comparison algorithms were:**Bitrate streaming** [24]: Download a GoP only when it needs to be played back, with UE running mode always in the LTE-ACTIVE mode. VLC uses this strategy by default.**Energy-aware** [10]: Switch between ON-OFF streaming and bitrate streaming depending on whether the user touches the screen. If the user touches the screen, the strategy realizes that the user tends to skip/quit and uses bitrate streaming. Otherwise, ON-OFF streaming is used to save energy.**Our method**: Consider the impact of the physical environment on user behavior by predicting the user’s watching state within a defined window. If it is unstable, use bitrate streaming; otherwise, use ON-OFF streaming to minimize energy consumption and unused data.

The downloading experiments were performed on the dataset mentioned in Section 3. The characteristics of the test videos are listed in Table 3. We used bitrate streaming, the energy-aware approach, and our solution to watch these videos in three cases of user operation. In the first case, there was no skipping during the entire video watching process. In the second case, several skips were performed during video watching. The operation mode was determined based on the sequence of operations in the test dataset. In the last case, parts of videos were skipped, and the screen was touched (without skipping/quitting) during the video watching process. To evaluate the impact of environmental factors on user operation, two physical environments (quiet and noisy) were considered; therefore, three sets of experiments were performed.

#### 5.2.1. Experiments with a Single Video in a Single Environment

For the three algorithms, when the operations on a single video were performed in the same environment, the results were as shown in Figure 3 separately for the three cases. The video “Java language features” was used in this part.

**Without skipping**: No operations or screen touches occurred during the entire video watching process. Figure 3a shows the energy consumption of various algorithms. We observe that the bitrate streaming algorithm had the highest energy consumption because the UE running mode was always the LTE-ACTIVE mode. The energy-aware algorithm had the lowest energy consumption since the screen was not touched, and the user was always in the stable state during playback. Therefore, the ON-OFF strategy was adopted at all times to save energy, saving 70.49% compared to bitrate streaming. For our algorithm, the amount of energy consumption depended entirely on the accuracy of the prediction results of the user watching state. Because our prediction accuracy was not 100%, the prediction result would change the watching state so that the switching download strategy could slightly increase the energy consumption compared to that of the energy-aware algorithm.

**With skipping**: We took a sample of the operation sequence of the courseware from the test dataset; the operation sequence was represented as a series of operation(time). The test sample was 1(29 October 2016 16:22:16) 4(22:21) 4(22:59) 4(23:01) 4(23:04) 4(23:05) 4(23:07) 4(23:39) 4(23:40) 4(23:42) 4(23:43) 4(23:47) 4(23:49) 4(23:53) 4(23:54) 4(23:56) 4(23:57) 4(24:00) 4(24:01) 4(24:39) 4(24:40) 1(36:32) 3(40:02) 1(40:42) 2(42:06). There was no further screen touching during the playback process, and this sequence of operations during playback was analyzed under various algorithms. The results are shown in Figure 3b,c. From the figure, we observe that the energy consumption of bitrate streaming was lower than that of (a), but higher than that of the other two methods. The power consumption figures of the energy-aware algorithm and our method were very similar, but our method downloaded less data because the energy-aware algorithm downloaded data to the buffer until the buffer was filled when no screen touching occurred. In contrast, our method downloaded only $D_{t}$ in the ON-OFF mode, resulting in a reduction in the total amount of downloaded data.

**With skipping and screen touching**: We used the previous sequence of operations and added a screen touch between every two operations. The experimental results are shown in Figure 3d,e. From Figure 3d, we observe that bitrate streaming remained stable compared with the second case. However, according to the energy consumption, our scheme outperformed the energy-aware algorithm by 19.14%. In this case, screen touching was added, and the download time of bitrate streaming would increase for the energy-aware algorithm; however, the video was not skipped at this time.

The experiment showed that, when there were many screen touches during the video watching period, our strategy obtained the lowest energy consumption and the mobile data usage. When there were few user operations, the energy savings of our approach would be reduced, but the data usage remained less than that of the benchmarks.

#### 5.2.2. Experiments with Different Videos in a Single Environment

For two videos with different durations (“Java language features” and “thread scheduling”), experiments were performed in the same environment (quiet) using the three algorithms. The operation trace of “thread scheduling” was 1(8 March 2017 15:45:34) 4(15:47:22) 4(15:49:43) 2(15:52:35), and the new results are shown in Figure 4 for the three cases, which can be compared with the counterparts in Figure 3.

**Without skipping**: For all three methods, when the frame rate and resolution were constant, the power consumption increased dramatically with the increase in video duration, as can be seen from Figure 3a and Figure 4a.

**With skipping**: Comparing the energy-aware algorithm and our algorithm with bitrate streaming in Figure 3b and Figure 4b, respectively, it can be seen that watching “thread scheduling” saved more power and wasted less downloaded data than watching “Java language features”. This result is mainly because the duration of the former video was shorter, resulting in a smaller proportion of skip operations and a larger portion of the video being viewed, compared to that of the “Java language features” video; thus, the number of switches of the download strategy to bitrate streaming was reduced, resulting in more significant energy savings.

**With skipping and screen touching**: As the proportion of screen touches and skip operations increased, the power consumption of the energy-aware algorithm increased relative to that in the **with skipping** case, while our method showed little increase.

#### 5.2.3. Experiments with a Single Video in Different Environments

For “Java language features”, experiments were performed in quiet and noisy environments using the three algorithms. For the noisy environment, the user operation trace from the Section 3 dataset was 1(31 October 2016 20:54:04) 4(20:54:00) 4(20:54:06) 4(20:54:07) 4(20:54:08) 4(20:54:08) 4(20:54:09) 4(20:54:10) 4(20:54:11) 4(20:54:12) 4(20:54:14) 4(20:54:15) 4(20:54:17) 4(20:54:18) 4(20:54:21) 4(20:54:23) 4(20:54:25) 4(20:54:25) 4(20:54:26) 4(20:54:28) 4(20:54:29) 4(20:54:29) 4(20:54:29) 4(20:54:31) 4(20:54:32) 4(20:54:33) 4(20:54:34) 4(20:54:35) 4(21:03:32) 4(21:03:33) 4(21:03:34) 4(21:03:36) 4(21:03:38) 4(21:03:38) 2(21:04:28). The results are shown in Figure 3 and Figure 5 for the three cases.

Energy-aware and our method were in touch screen and unstable states for a long time under both **with skipping** and **with skipping and screen touching**; thus, they saved less power than in a quiet environment, as shown in Figure 3b,d and Figure 5a,c. The reason was assumed to be the increase in skipping operations for the trace in the noisy environment. To determine statistically the effect of the physical environment on user operation, we separately counted the number of skip operations users performed in quiet and noisy environments. As shown in Figure 6, skipping operations were performed more often in the noisy environment (red dots) than in the quiet environment (green dots).

In the scenario of a single video and multiple environments, since different physical environments were ultimately reflected in skip times during the viewing process, the result was the same as before; i.e., the more skips there were, the lower the energy savings.

## 6. Conclusions

In this paper, we proposed a mobile video streaming scheme to reduce energy consumption and bandwidth usage in an LTE network. A prediction model of the user watching state was developed using machine learning, considering the influence of the user’s physical environment on user behavior. Afterwards, a download algorithm was introduced based on the UE running mode in the LTE system. The download strategies were adjusted according to the user watching state. When the state was stable, the UE running mode was LTE-IDLE for as long as possible to save energy, and the ON-OFF streaming strategy was used. When the state was unstable, the skip/quit operations occurred. The bitrate streaming strategy was used to minimize the download of unused data. The scheme was implemented in the VLC player on the Android platform, and the performance of the scheme was compared with that of the existing methods under three scenarios of video watching. The experimental results showed that our method was effective and performed best when the user touched the screen frequently.

Considering the physical environment, this paper studied on-demand downloading of online video data, mainly concerning MP4 file transfer under the HTTP/1.1 protocol, in which data are actively requested according to user’s needs. Recently, delivery-level research on HAS over HTTP/2 has received increasing attention. Switching between the ON-OFF streaming strategy and the bitrate streaming strategy is also suitable for HTTP/2-based adaptive streaming. The switching strategy can be applied to the dynamic server push strategy to determine the push number of HTTP/2-based adaptive streaming. Therefore, future research will focus on how to provide cost-effective HTTP/2-based adaptive streaming through the environment-aware watching state prediction. In the meantime, detailed video information affecting user’s viewing state prediction, such as the way of video production, the difficulty of video courseware, and the chroma characteristics of the video, was not taken into account in our research. It can be analyzed and added in the follow-up study to achieve more accurate user watching state prediction.

## Figures and Tables

**Figure 1 sensors-19-03654-f001:**
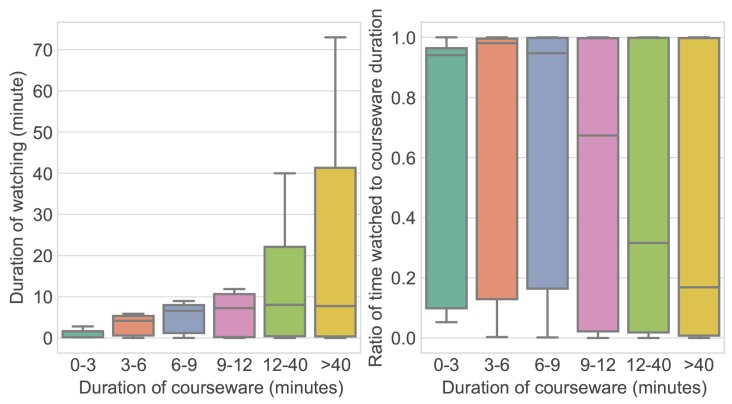
Distribution of a user’s watching time and the ratio of watching time to video duration for various durations of video courseware.

**Figure 2 sensors-19-03654-f002:**
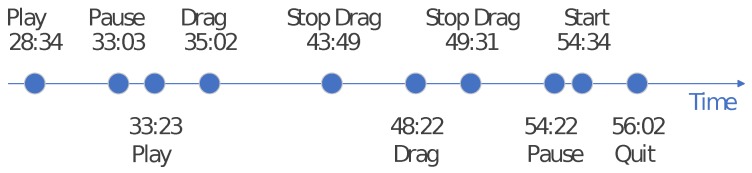
Trace of a video watching process.

**Figure 3 sensors-19-03654-f003:**
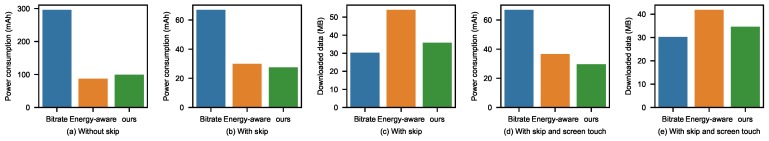
Amount of power consumed and downloaded data size for “Java language features” under three different methods for a single video in the same environment: (**a**,**b**,**d**) show the power consumption in the three cases, while (**c**,**e**) illustrate the corresponding downloaded data size.

**Figure 4 sensors-19-03654-f004:**
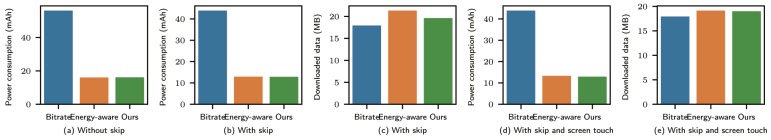
Amount of power consumed and downloaded data size under three different methods for “thread scheduling” in the same environment: (**a**,**b**,**d**) show the power consumption in the three cases, while (**c**,**e**) illustrate the corresponding downloaded data size.

**Figure 5 sensors-19-03654-f005:**
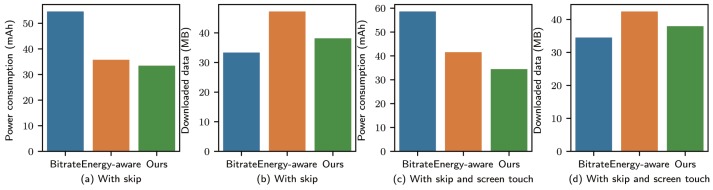
Amount of power consumed and downloaded data size under three different methods for “thread scheduling” in the same environment: (**a**,**b**,**d**) show the power consumption in the three cases, while (**c**,**e**) illustrate the corresponding downloaded data size.

**Figure 6 sensors-19-03654-f006:**
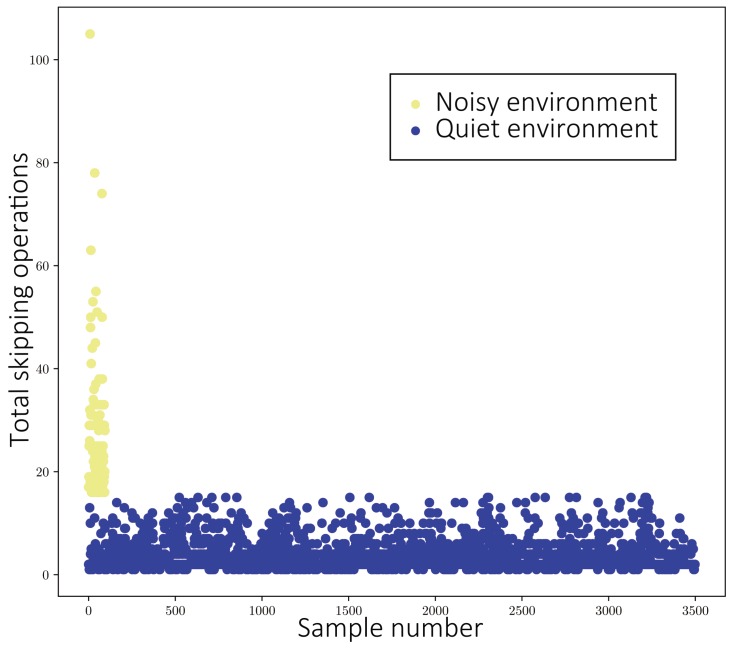
The distribution of skipping operations for quiet and noisy environments.

**Table 1 sensors-19-03654-t001:** Final representation of semantics with n=4,k=50.

Semantic	4-Digit *Patterns*
Stable	1313, 1312, 1342, 1432,
Unstable	1344, 4444, 4442, 4443, 4431, 3444,

**Table 2 sensors-19-03654-t002:** Modeling results under 7 machine learning algorithms.

Modeling Method	Modeling Time (s)	$R^{2}$	MAE	RMSE
Ridge	3.66	0.6108	0.0122	0.0820
Lasso	0.90	0.5813	0.0181	0.0823
Elastic Net	4.49	0.6512	0.0152	0.0814
XGBoost	898.41	0.7886	0.0269	0.0907
Random Forest	12.76	0.8332	0.0093	0.0114
ExtraTrees	1.59	0.8183	0.0151	0.0123
Gradient Boosting	9.26	0.8080	0.0175	0.0856

**Table 3 sensors-19-03654-t003:** Test videos.

Video Name	Resolution	Frame Rate	Duration	Size
Java Language Features	960 × 720	25	2583 s	121 MB
Thread Scheduling	960 × 720	25	483 s	22.4 MB

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
