# Peer review of "Toward Cost-Effective Mobile Video Streaming through Environment-Aware Watching State Prediction"

_sensors, 2019, doi:10.3390/s19173654_

Round 1

Reviewer 1 Report

This paper proposes a mobile video streaming scheme that considers both the physical environment and the operation behavior of a mobile user.

1) A prediction model of the user watching state is developed using ML. This prediction model reveals the influence of the user’s physical environment on the user’s operation behavior.

2) A downloading algorithm provides differentiated download strategies according to stable and unstable user watching states. When the state is stable, the video streaming scheme saves energy through data transmission batching. When it is unstable, the video streaming scheme minimizes the download of unused data via a conservative download method.

3) On the Android platform, the authors developed a mobile video streaming prototype based on the VLC player.

I consider that this work has merit because the proposed scheme can effectively reduce power consumption and mobile data usage. This evidence is based on the experimental results presented. However, to make this paper acceptable the authors need to respond to the following substantive points.

 CONSTRUCTIVE COMMENTS

The “flow” of the ideas presented in the paper must be improved. High quality arguments are required in Section “4. Downloading algorithm based on the watching state.” For example, the authors must state the limitations of the downloading algorithm. In Figure 7, noisy and quiet environments should be depicted with non similar (very discrete) colours. Some Refs should be completed and corrected. In the section 2. "Related work", the authors should correct the way they cite their Refs. For example, in line 64, the should write "Moreover, Ghoreishi et al. [4].." English language should be improved/polished.

Reviewer 2 Report

The paper is very well organized and clear. The contributions are interesting. The impact of the physical environment on user behavior is taken into account to predict user behavior and improve video transmission.

Sensor data and learning operation logs includes data for 3905 students and 380 courses (6164 courseware videos). In total, 57207 samples were obtained with represents a very interesting dataset.

User state is divided into two categories: stable and unstable. Is this case, a confusing matrix could be presented. It would help in the understanding of the quality of the prediction.

User state is divided into two categories: stable and unstable. Can the authors argue why only two states? The employment of more states could help or not the system?

Figure 3 is quite unusual. Machine learning results often are presents in the form of a boxplot, with results from 30 runs for each methods. Please have a look in Figure 8 of the paper "A Machine-Learning Approach to Distinguish Passengers and Drivers Reading While Driving" Sensors 2019. Use the figure as an example to improve your presentation. Also, for the prediction system based on machine learning, please add a statistical comparison among the methods.

Please describe how the physical environment data affects the prediction. It would be very useful for understanding witch environmental data is more important to predict user state.
